# QuaMo: Quaternion Motions for Vision-based 3D Human Kinematics Capture

**Cuong Le**[1]**, Pavlo Melnyk**[1]**, Urs Waldmann**[1]**, Mårten Wadenbäck**[1]**, Bastian Wandt**[2]
[1] Linköping University, Sweden
[2] Independent researcher
`cuong.le@liu.se`

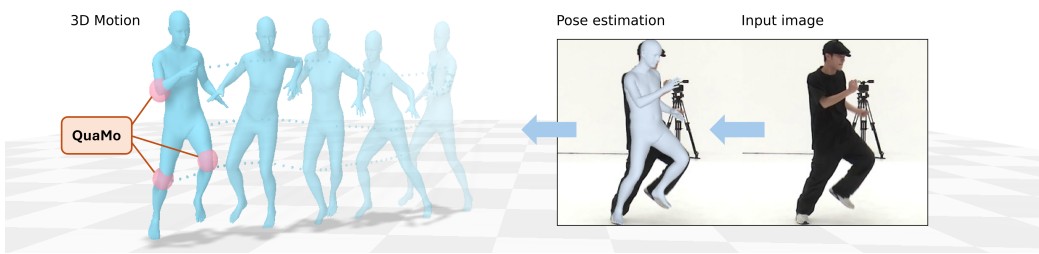

Figure 1: We present **QuaMo**, a novel online 3D human kinematics capture approach based on **Qua**ternion **Mo**tions (pink), modeled via a meta-PD algorithm with acceleration enhancement. Given a vision-based 3D pose estimation as prior, QuaMo predicts plausible and accurate motions.

## Abstract

Vision-based 3D human motion capture from videos remains a challenge in computer vision. Traditional 3D pose estimation approaches often ignore the temporal consistency between frames, causing implausible and jittery motion. The emerging field of kinematics-based 3D motion capture addresses these issues by estimating the temporal transitioning between poses instead. A major drawback in current kinematics approaches is their reliance on Euler angles. Despite their simplicity, Euler angles suffer from discontinuity that leads to unstable motion reconstructions, especially in online settings where trajectory refinement is unavailable. Contrarily, quaternions have no discontinuity and can produce continuous transitions between poses. In this paper, we propose QuaMo, a novel Quaternion Motions method using quaternion differential equations (QDE) for human kinematics capture. We utilize the state-space model, an effective system for describing real-time kinematics estimations, with quaternion state and the QDE describing quaternion velocity. The corresponding angular acceleration is computed from a meta-PD controller with a novel acceleration enhancement that adaptively regulates the control signals as the human quickly changes to a new pose. Unlike previous work, our QDE is solved under the quaternion unit-sphere constraint that results in more accurate estimations. Experimental results show that our novel formulation of the QDE with acceleration enhancement accurately estimates 3D human kinematics with no discontinuity and minimal implausibilities. QuaMo outperforms comparable state-of-the-art methods on multiple datasets, namely Human3.6M, Fit3D, SportsPose and AIST. The code is available at ⊙.

## 1 Introduction

Monocular 3D human motion capture is a challenging problem in computer vision due to the loss of depth information and the complexity of body articulation. Traditional 3D human pose estimation (HPE) approaches, which directly estimate 3D joint positions or angles, achieve high accuracy on distance-based evaluation metrics (Pavllo et al., 2019; Zheng et al., 2021; Kocabas et al., 2020; Sun et al., 2023; Goel et al., 2023). However, when considering a captured trajectory over consecutive

frames from a video, 3D HPE often results in implausible artifacts such as jittery or unnatural poses. Addressing these challenges by introducing physics models (*i.e.*, state-space model with velocity and acceleration) to enforce temporal consistency between consecutive poses is an emerging research direction, fusing vision-based predictions with human kinematic chains (Shimada et al., 2020) or volumetric models (Loper et al., 2015). Our proposed method, QuaMo, falls into this category.

The foundation of all kinematics-based approaches is the nonlinear state-space model, where human poses and their corresponding velocities are computed from either meta-PD controllers with learnable PD gains (Shimada et al., 2021; Li et al., 2022a; Le et al., 2024a), physics simulation engines (Yuan et al., 2021; Gärtner et al., 2022b;a), or neural networks (Rempe et al., 2021). Using state-space modeling, the process of predicting the next human poses is equivalent to solving a time-series ordinary differential equation (ODE) (Chen et al., 2018). Our proposed method, QuaMo, serves as the function that takes the current human pose as input and predicts the corresponding velocity, giving an estimate for the next pose. We develop QuaMo as an online approach, only relying on the single time step input, making QuaMo applicable to real-time applications (autonomous driving (Priisalu et al., 2020; Wang et al., 2024), or biomechanics (Bogert et al., 2013; Uhlrich et al., 2023)).

Current modern temporal-based approaches often opt for Euler angles (Yuan et al., 2021; Rempe et al., 2021; Li et al., 2022a) as the main joint orientation representation for human kinematics estimation. Despite their simplicity and intuitive interpretation, Euler angles have two well-known issues: singularities (a.k.a. gimbal lock) and discontinuities (at angles $0$ and $2\pi$) (Shoemake, 1985). Discontinuities cause the joint to incorrectly rotate backwards to the intended direction, resulting in highly unstable motion reconstructions. Quaternions are known to resolve the discontinuity problem by representing orientations with a four-dimensional vector, but have not received proper studies within the field. To this end, we propose using quaternion joints for kinematics estimation, with their velocity computed from a novel acceleration enhanced PD control. Unlike Euler angles, the quaternion derivative cannot be approximated by a finite difference between respective elements due to rotational constraints (Kuipers, 1999) – we use an operation based on the Hamilton product.

The underlying state-space model of QuaMo consists of a joint orientation represented as a quaternion and an angular velocity state, resulting in two main parallel streams (Fig. 2): 1) a quaternion first-order derivative calculation based on the Hamilton product between the current quaternion and the newly computed angular velocity, and 2) a meta-PD algorithm with newly developed acceleration enhancement to estimate the rotational velocity derivative. Unlike prior work (Shimada et al., 2021; Le et al., 2024a), we apply the exact quaternion integration solution under unit sphere constraint, eliminating any approximation errors that arise when using the traditional Euler integration (a.k.a. first-order Runge–Kutta) method (Andrle & Crassidis, 2013). The novel acceleration enhancement term, computed based on the second-order quaternion difference between reference poses, adaptively compensates the signals of the PD algorithm for more accurate kinematics estimates. Specifically, the acceleration term increases the control signals when sudden pose changes occur (fast movement) and dampens the signals upon reaching the target poses. A demonstration of our proposed approach can be seen in Fig. 1.

Our proposed approach is evaluated against current state-of-the-art kinematics-based methods on the Human3.6M (Ionescu et al., 2014), Fit3D (Fieraru et al., 2021), SportsPose (Ingwersen et al., 2023), and a subset of AIST (Li et al., 2021b) datasets. In summary, our main contributions are:

- We propose a quaternion differential equation with quaternion as joint rotation for 3D human motion estimation, inherently overcoming the drawbacks of Euler angles.
- We introduce a novel acceleration enhancement that adaptively regulates the angular acceleration based on quick movement changes for more accurate motion estimation.
- We show that solving the QDE under the quaternion unit-sphere constraint $\mathcal{S}^3$ results in more plausible and accurate human poses in online real-time settings.

## 2 RELATED WORK

**3D Human Pose Estimation**. Traditionally, 3D human motion capture is addressed via 3D pose estimations, either 1) lifting from 2D cues (Wang et al., 2019; Pavllo et al., 2019; Wandt & Rosenhahn, 2019; Li et al., 2020; Xu et al., 2020; Wandt et al., 2021; Zheng et al., 2021; Zhao et al., 2023; Peng et al., 2024; Cai et al., 2024; Sun et al., 2024; Lang & Chuah, 2025; Huang et al., 2025; Kim et al., 2025), or 2) directly estimating 3D human poses from input images (Pavlakos et al., 2017;

Kocabas et al., 2020; Li et al., 2021a; 2022b; You et al., 2023; Wang & Daniilidis, 2023; Kim et al., 2023; Baradel et al., 2024; Dwivedi et al., 2024; Le et al., 2024b; Patel & Black, 2025). Despite the low average joint error, methods lifting from 2D do not consider the human body constraint, *i.e.* bone-length consistency between consecutive 3D poses, thus cannot be compared to template-based approaches, and this argument has been raised by related work (Gärtner et al., 2022a; Li et al., 2022a; Zhang et al., 2024). To impose natural body constraints, modern approaches utilize volumetric human models such as SMPL (Loper et al., 2015) as a prior and only estimate the angular poses for the models, fitting them to 2D and 3D observations (Pavlakos et al., 2019; Kanazawa et al., 2019; Mahmood et al., 2019; Zhu et al., 2023). This line of research often overlooks the temporal consistency between consecutive estimated poses, leading to implausible artifacts such as jittery, foot-skating, and unnatural poses. In this work, we address these issues by employing a kinematics-based approach, taking into account the temporal consistency between consecutive 3D estimations.

**3D Human Kinematics Capture**. Unlike 3D HPE, human kinematics approaches enforce temporal consistency to eliminate motion artifacts created by monocular estimation, either through pose priors (Huang et al., 2022; Rempe et al., 2021), or physics laws and constraints (Shimada et al., 2020; Gärtner et al., 2022a; Li et al., 2022a; Tripathi et al., 2023; Zhang et al., 2024).

Trajectory optimization is a popular approach for kinematics-based 3D human motion capture (Al Borno et al., 2013; Shimada et al., 2020; Rempe et al., 2020; Xie et al., 2021; Gärtner et al., 2022b;a). These approaches commonly introduce kinematics constraints in the form of physics laws as their main optimization objective. This poses a challenge where the physical constraints are required to be differentiable and the optimization is often done offline. Recent approaches extend the physics modeling with comprehensive contact estimations from modern physics engines Coumans & Bai (2016–2019); Todorov et al. (2012). However, these contact models are non-differentiable, motivating a subfield of motion imitation research that utilizes reinforcement learning with physics constraints in their reward designs (Yu et al., 2021; Yuan et al., 2021; Peng et al., 2022; Yao et al., 2022; Huang et al., 2022; Yuan et al., 2023). A major problem with trajectory optimization and reinforcement learning is the limited adaptation to unseen motions.

Recently, learning 3D human kinematics from data has received more attention, thanks to its strong generalization capability. Rempe et al. (2021) use a conditional variational autoencoder to generate the next human pose, implicitly treating the latent variables as motion kinematics. While the latent kinematics is learned, Rempe et al. (2021) still require a test time optimization process to refine their estimation. In contrast, leveraging off-the-shelf monocular 3D HPE (Kocabas et al., 2020; Li et al., 2022b) as the targets for motion reconstruction can alleviate the offline optimization while retaining robust and physically plausible kinematics estimation. Zhang et al. (2024) utilize a transformer-based autoencoder that takes the full sequence of monocular 3D poses as inputs, predicts the corresponding sequence while enforcing physics constraints in the latent space. To maintain the explicit temporal consistency between frame-wise predictions, Shimada et al. (2021) introduces the usage of a meta-PD controller for predicting the motion dynamics based on the 2D pose cues estimated from video. While the 3D kinematics capture works online, Shimada et al. (2021) still require a pre-filtering of 2D cues to ensure plausible 3D estimations. Li et al. (2022a) apply the PD controller with temporal convolutions and an attentively refined target pose from the full sequence for robust motion estimation. The key similarity between these approaches is the access to future monocular cues, preventing them from real-time deployments. Le et al. (2024a) address the imperfection of *online* PD-based simulation by re-integrating the input poses into the final prediction via a learnable Kalman filter. However, re-introducing noisy kinematics has the potential to break the temporal consistency enforced by the integration scheme. Furthermore, one source of error for poor simulations is the usage of Euler angles that are prone to representation changes when enforcing temporal consistency frame-wise (Allgeuer & Behnke, 2018; Yang, 2019).

Our work contributes towards the online 3D human kinematics capture, using a meta-PD algorithm with the robust quaternions as the joint orientation representation. The kinematics estimation follow the correct Lie-group constrained calculation for quaternions, while additionally dampened by a novel second-order control compensation that results in smoother and more accurate motions.

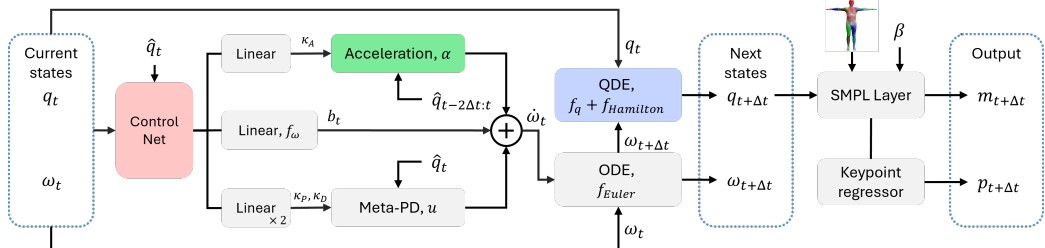

Figure 2: QuaMo consists of two differentiable equations: ODE for angular velocity $\omega$ and QDE for quaternion pose $q$. The updated $\omega_{t+\Delta t}$ is computed via a data-driven meta-PD controller with the additional adaptive signals from our novel second-order acceleration enhancement and Euler integration. Given $\omega_{t+\Delta t}$, the next human pose $q_{t+\Delta t}$ is updated by solving the QDE with the Hamilton quaternion product. The human body mesh $m_{t+\Delta t}$ and the corresponding keypoints $p_{t+\Delta t}$ are retrieved by applying a linear transformation with the SMPL skinned model from Pavlakos et al. (2019), taking the pose $q_{t+\Delta t}$ and shape parameter $\beta$ as inputs.

## 3 METHODOLOGY

### 3.1 OVERALL PIPELINE

We model human motion via a state-space system. Given $N$ joints in the human body, let $Q \in \mathbb{R}^{N \times 4}$ be the human pose tensor consisting of joint relative rotations represented as $N$ quaternions $q \in \mathbb{H}$ (the Hamilton set). The corresponding angular velocities are denoted as $\omega \in \mathbb{R}^3$. We use the SMPL body model from Loper et al. (2015) with $N = 24$ body joints with the root rotation at the first entry. The discrete-time state-space model of the system with a sampling rate of $\Delta t$ is

$$\begin{bmatrix} \omega_{t+\Delta t} \\ q_{t+\Delta t} \end{bmatrix} = \begin{bmatrix} f_{\text{Euler}}(\omega_t, \dot{\omega}_t, \Delta t) \\ f_{\text{Hamilton}}(q_t, \dot{q}_t, \Delta t) \end{bmatrix}, \begin{bmatrix} \dot{\omega}_t \\ \dot{q}_t \end{bmatrix} = \begin{bmatrix} f_\omega(q_t, \omega_t) + u(q_t, \omega_t, \hat{q}_t) + \alpha(\hat{q}_{t-2\Delta t:t}) \\ f_q(q_t, \omega_{t+\Delta t}) \end{bmatrix}, \quad (1)$$

where $\dot{q}_t$ and $\dot{\omega}_t$ are the first-order derivatives of $q$ and $\omega$ at time step $t$, which can also be referred as the quaternion velocity and angular acceleration; $q_{t+\Delta t}$ and $\omega_{t+\Delta t}$ are the pose and angular velocity at the next time step $t + \Delta t$. The function $f_{\text{Euler}}$ estimates the next-step angular velocity via Euler integration. The function $f_{\text{Hamilton}}$ estimates the next-step quaternion pose using Hamilton operations. The angular acceleration $\dot{\omega}_t$ is modeled via three functions: 1) a data-driven $f_\omega$ that directly predicts the acceleration given the current states $q_t$ and $\omega_t$; 2) an external control signal $u$ based on the meta-PD algorithm given the reference pose $\hat{q}_t$ from an off-the-shelf 3D pose estimator; and 3) a novel acceleration enhancement term $\alpha$ computed from the last three reference poses $\hat{q}_{t-2\Delta t:t}$. The quaternion velocity $\dot{q}_t$ rotates the current pose $q_t$ to $q_{t+\Delta t}$ along the unit sphere $\mathcal{S}^3$ group of quaternion multiplication (Andrle & Crassidis, 2013). Fig. 2 shows how we build our pipeline with these functions. The predicted pose $q_{t+\Delta t}$ is used to control an SMPL model together with a shape parameter $\beta \in \mathbb{R}^{10}$, resulting in a human mesh $m_{t+\Delta t} \in \mathbb{R}^{6890 \times 3}$. We apply a joint regressor to obtain the keypoint pose $p_{t+\Delta t} \in \mathbb{R}^{17 \times 3}$, *i.e.* the 17 keypoints from Ionescu et al. (2014), or the COCO 17 keypoints from Lin et al. (2014).

### 3.2 QUATERNION DIFFERENTIAL EQUATION

**A quaternion** $q \in \mathbb{H}$ is represented by $(q_0, q_1, q_2, q_3) \in \mathbb{R}^4$, also written as:

$$q = q_0 + q_1 i + q_2 j + q_3 k , \quad (2)$$

where $i$, $j$, and $k$ are imaginary units satisfying $i^2 = j^2 = k^2 = ijk = -1$. The sum of two quaternions $p$ and $q$ is obtained by summing their respective scalar coefficients: $p + q = (p_0 + q_0) + (q_1 + p_1)i + (q_2 + p_2)j + (q_3 + p_3)k$. The (Hamilton) product $\otimes$ of two quaternions is defined as $p \otimes q = (p_0 q_0 - p^\top q, \ p_0 q + q_0 p + p \times q)$, with the crucial non-commutative property $p \otimes q \neq q \otimes p$. The non-commutativity of the Hamilton product is key to *unit* quaternions, *i.e.* quaternions normalized to unit length, $\|q\| = 1$, being broadly used to represent 3D rotations, with a number of advantages over other representations, *e.g.*, gimbal lock and discontinuity avoidance. For a unit quaternion $q \in \mathcal{S}^3$, $q_0 = \cos(\alpha/2)$ is the scalar part and $(q_1, q_2, q_3) = e \sin(\alpha/2)$ is the vector part representing a rotation about axis $e \in \mathbb{R}^3$ by angle $\alpha$.

**Quaternion derivative** over time, $\dot{q}$, is an important concept in various fields including robotics, physics, and engineering (*e.g.*, in spacecraft modeling Yang (2019); Fresk & Nikolakopoulos (2013); Gołąbek et al. (2022)). Given the angular velocity $\omega = (\omega_1, \omega_2, \omega_3) \in \mathbb{R}^3$, the corresponding quaternion velocity is defined via the following *quaternion differential equation* (QDE):

$$\dot{q} = \frac{1}{2}\Omega(\omega)q = \frac{1}{2}\begin{bmatrix} -[\omega]_\times & \omega \\ -\omega^\top & 0 \end{bmatrix} q\,, \quad \text{where } [\omega]_\times = \begin{bmatrix} 0 & -\omega_3 & \omega_2 \\ \omega_3 & 0 & -\omega_1 \\ -\omega_2 & \omega_1 & 0 \end{bmatrix}. \tag{3}$$

In this work, the QDE effectively describes the quaternion transitioning between 3D human poses without any representation discontinuity. Assuming constant angular velocity $\omega_t$ during $\Delta t$, the quaternion pose solution to Eq. 3 at time step $t + \Delta t$ can be written as

$$q_{t+\Delta t} = \exp\left(\frac{\Delta t}{2}\Omega(\omega_{t+\Delta t})\right) q_t = q_\omega \otimes q_t\,, \tag{4}$$

where $\exp(\frac{\Delta t}{2}\Omega(\omega))$ is the rotation matrix, that rotates $q_t$ using $\omega_t$. Equivalently, the next pose $q_{t+\Delta t}$ can be obtained by the Hamilton product between $q_\omega$, the quaternion representation of the rotation matrix, and $q_t$. Compared to the integration approximation in prior work that violates the quaternion constraint of $\mathcal{S}^3$ (Supplementary E), integration by Eq. 3 and Eq. 4 ensures an exact quaternion solution at all times, leading to more accurate estimations, as demonstrated via the ablation in Tab. 3.

### 3.3 META-PD CONTROLLER WITH SECOND-ORDER ACCELERATION

The *ordinary differential equation* (ODE) that describes the motion acceleration $\dot{\omega}_t$ is written as

$$\dot{\omega}_t = \underbrace{\kappa_P\big(\text{vec}(\hat{q}_t \otimes q_t^*)\big) - \kappa_D\,\omega_t}_{\text{meta-PD algorithm}} + \underbrace{b_t}_{\text{bias}} + \underbrace{\kappa_A\big(\text{vec}(\hat{q}_t \otimes \hat{q}_{t-\Delta t}^*) - \text{vec}(\hat{q}_{t-\Delta t} \otimes \hat{q}_{t-2\Delta t}^*)\big)}_{\text{acceleration enhancement}}, \tag{5}$$

where $\kappa_P \in \mathbb{R}$ and $\kappa_D \in \mathbb{R}$ constitute the PD controller's proportional-derivative gains, while $\kappa_A \in \mathbb{R}$ is the scaling factor of the newly introduced second-order acceleration. The control signals $b_t$, $\kappa_P$, $\kappa_D$, $\kappa_A$ are predicted by a *ControlNet* via linear projections from the latent embedding, given $q_t$, $\omega_t$ and $\hat{q}_t$ as inputs (Fig. 2). Inspired by Fresk & Nikolakopoulos (2013), the meta-PD controller is computed proportionally to the vector, *i.e.* imaginary, part of the quaternion error between $\hat{q}_t$ and the complex conjugate of $q_t$, $q_t^*$. Due to the online setting considered in our work, the estimated $\hat{q}_t$ is inherently noisy, especially when obtained from a direct regression method such as Sun et al. (2023). We suggest the derivative term $\kappa_D\,\omega_t$ to dampen the predicted proportional control signal $\kappa_P\big(\text{vec}(\hat{q}_t \otimes q_t^*)\big)$, effectively reducing overshooting and jittery. The data-driven $b_t$ is the estimation of $f_\omega$ approximated from the embedding of *ControlNet*, commonly referred as bias term in prior work (Shimada et al., 2020; Li et al., 2022a).

The proposed **acceleration enhancement** term is the best guess of the person's intended target pose, computed from the second-order quaternion difference of the last three reference poses $\hat{q}_{t-2\Delta t:t}$, resulting in the angular acceleration of the reference signal. This term, scaled by $\kappa_A$, reacts adaptively to the rate of change in reference signals, *i.e.*, it positively reinforces the controller when the reference $\hat{q}$ changes fast (quick movements to reach a new target) and dampens the control signals as the motion moves closer to the target. The adaptive nature of the acceleration enhancement helps the kinematics motion react quickly to the intended target, while maintaining minimal overshooting.

**Global translation**. We also compute the root translation $r_t$ with meta-PD and Euler integration (Li et al., 2022a). The calculation is written as: $r_{t+\Delta t} = r_t + (v_t + (\kappa_P(\hat{r}_t - r_t) - \kappa_D\,v_t)\Delta t)\Delta t$, with $\hat{r}_t$ as the reference root position and $v_t$ as the current root linear velocity. The global motion trajectory is then obtained by adding the translation $r_t$ to the body mesh $m_t$ or keypoints $p_t$ respectively. A stability analysis of global translation can be found in Supplementary F.

### 3.4 TRAINING OBJECTIVES

The total objective $\mathcal{L}_{\text{total}}$ for capturing 3D human motion consists of three different loss terms, defined in Eq. 6. The first loss is the local reconstruction loss $\mathcal{L}_{\text{local}}$, which is the frame-wise L1 distance between the predicted root-aligned 3D keypoint $p_t$ to the ground truth $p_t^{GT}$ from the respective dataset. The same calculation applies to the root translation $r_t$. The second loss is the

global consistency loss $\mathcal{L}_{\text{global}}$, which is the average L1 distance between second-order finite differences of the predicted motion $p_{1:T}$ and ground truth $p_{1:T}^{GT}$. The finite differences are computed as $\ddot{x}_{0:T} = x_{0:T-2} - 2x_{1:T-1} + x_{2:T}$, with $x$ being either $p$ or $r$. We additionally fine-tune the shape parameter $\beta$ via a learnable $\beta_{\text{fix}}$. To prevent unrealistic body shapes, we introduce a regularization on $\beta_{\text{fix}}$, ensuring the estimated body shapes do not deviate far from the average human shape ($\beta = 0$).

$$\mathcal{L}_{\text{total}} = \mathcal{L}_{\text{local}} + \mathcal{L}_{\text{global}} + \lambda \mathcal{L}_{\text{beta}}, \quad \mathcal{L}_{\text{beta}} = \|\beta_{\text{fix}}\|,$$
$$\mathcal{L}_{\text{local}} = \frac{1}{T}\frac{1}{N}\sum_{}^{T}\sum_{}^{N}|p_{0:T}^{GT} - p_{0:T}| + \frac{1}{T}\sum_{}^{T}|r_{0:T}^{GT} - r_{0:T}|, \quad (6)$$
$$\mathcal{L}_{\text{global}} = \frac{1}{T}\frac{1}{N}\sum_{}^{T}\sum_{}^{N}|\ddot{p}_{0:T}^{GT} - \ddot{p}_{0:T}| + \frac{1}{T}\sum_{}^{T}|\ddot{r}_{0:T}^{GT} - \ddot{r}_{0:T}|.$$

## 4 EXPERIMENTS

### 4.1 DATASETS

We evaluate QuaMo on four established motion capture datasets. The main dataset in comparison with related methods is Human3.6M (Ionescu et al., 2014). The dataset contains diverse human motion capture data from seven actors in a laboratory setup. Following prior work (Shimada et al., 2021; Yuan et al., 2021; Le et al., 2024a), data from the first five actors (S1, S5, S6, S7, S8) is used for training, while S9 and S11 are reserved for testing. For a fair comparison, as suggested by Shimada et al. (2021), only actions that involve foot-ground contacts are considered. To demonstrate the performance of our method on more diverse motions, we additionally evaluate on the Fit3D (Fieraru et al., 2021) and SportsPose dataset (Ingwersen et al., 2023). The former contains complex exercise motions with a laboratory setup similar to Human3.6M. The latter comprises sport action videos taken with a mobile phone in different scene setups. We employ the training split from Le et al. (2024a) for evaluation. Lastly, following Gärtner et al. (2022a), we test QuaMo on a subset of AIST (Li et al., 2021b) (details in Supplementary C), consisting of dancing videos with pseudo ground-truths from 3D triangulation.

### 4.2 IMPLEMENTATION DETAILS

QuaMo is implemented as an online end-to-end approach as in Fig. 2. At time step $t$, the system states $q_t$, $\omega_t$ and target pose $\hat{q}_t$ are used as inputs for the ControlNet, followed by two heads for $\kappa_P$ and $\kappa_D$, one head for data-driven bias $b_t$, and one head for $\kappa_A$ predictions of Eq. 5. The target poses $\hat{q}$ are initially extracted using TRACE (Sun et al., 2023) and HMR2.0 (Goel et al., 2023). Following (Shimada et al., 2021; Gärtner et al., 2022a; Le et al., 2024a), the extracted motions are down-sampled from 50Hz to 25Hz, first frame prediction root-aligned to the world origin, and then split into sub-sequences of 100 frames for batch training. The estimated pose $q$ is in SMPL format and uses a mesh-based linear regression to obtain the keypoint predictions. In addition to the ControlNet, we create an InitNet for creating the initial states $q_0$, $\omega_0$ and $\beta_{\text{fix}}$, taking only the first two target poses $\hat{q}_{0:1}$, and the first shape parameter $\beta_0$ (from either TRACE or HMR2.0) as inputs. Please refer to Supplementary A for details about InitNet and ControlNet.

For all experiments, QuaMo is trained for a total of 35 epochs with a batch size of 64 and an initial learning rate of $5e^{-4}$, with an exponential decay at epoch 20 and 30 by a factor of 10. The ControlNet consists of two fully connected layers with a hidden dimension of 512, followed by a *LayerNorm* and *LeakyReLU* activation. To stabilize the training in the beginning, in the first 5 epochs, the network parameters are updated per-frame with a learning rate of $1e^{-4}$ while the global loss $\mathcal{L}_{\text{global}}$ is turned off. During training, the shape loss $\mathcal{L}_{\text{beta}}$ is scaled by $\lambda = 0.01$. The time step $\Delta t = 0.04$ corresponds to the down-sampled motion capture rate of 25Hz. All experiments are reported with error bars obtained by testing on five different random seed values ($0 - 4$).

### 4.3 METRICS

We evaluate QuaMo on all of the datasets with two set of metrics: local and global. The local metrics consider the Mean Per Joint Position Error (MPJPE) (in mm) between root-aligned poses.

| Method | Tmpl. | Kin. | Onl. | Local metrics | | | Global metrics | | | |
|---|---|---|---|---|---|---|---|---|---|---|
| | | | | MPJPE | P-MPJPE | Accel | G-MPJPE | GRE | G-Accel | FS |
| PoseAnchor (Kim et al., 2025) | - | - | - | 40.3 | 32.1 | - | - | - | - | - |
| KTPFormer (Peng et al., 2024) | - | - | - | 40.1 | 31.9 | - | - | - | - | - |
| PoseMamba (Huang et al., 2025) | - | - | - | 37.1 | 31.5 | - | - | - | - | - |
| Mambapose Lang & Chuah (2025) | - | - | - | 36.5 | 28.6 | - | - | - | - | - |
| HMMR (Kanazawa et al., 2019) | ✓ | - | - | 79.4 | 55.0 | - | - | 231.1 | - | - |
| VIBE (Kocabas et al., 2020) | ✓ | - | - | 68.6 | 43.6 | 23.4 | 207.7 | - | - | 27.4 |
| MAED (Wan et al., 2021) | ✓ | - | - | 56.4 | 38.7 | - | - | - | - | - |
| HMR (Kanazawa et al., 2018) | ✓ | - | ✓ | 78.9 | 54.3 | - | - | 204.2 | - | - |
| IPMAN-R (Tripathi et al., 2023) | ✓ | - | ✓ | 60.7 | 41.1 | - | - | - | - | - |
| TRACE (Sun et al., 2023) | ✓ | - | ✓ | 56.1 | 39.4 | 18.9 | 143.0 | 127.2 | 39.4 | 80.3 |
| HybrIK (Li et al., 2021a) | ✓ | - | ✓ | 55.4 | 33.6 | - | - | - | - | - |
| MeshPose (Le et al., 2024b) | ✓ | - | ✓ | 50.7 | 35.4 | - | - | - | - | - |
| HMR2.0 (Goel et al., 2023) | ✓ | - | ✓ | 46.7 | 30.7 | 9.1 | 97.2 | 86.8 | 16.8 | 11.5 |
| PhysCap (Shimada et al., 2020) | ✓ | ✓ | - | 97.4 | 65.1 | - | - | 182.6 | - | - |
| TrajOpt (Gärtner et al., 2022b) | ✓ | ✓ | - | 84.0 | 56.0 | - | 143.0 | - | - | **4.0** |
| DiffPhy (Gärtner et al., 2022a) | ✓ | ✓ | - | 81.7 | 55.6 | - | 139.1 | - | - | 7.4 |
| Xie et al. (2021) | ✓ | ✓ | - | 68.1 | - | - | - | 85.1 | - | - |
| PhysPT (Zhang et al., 2024) | ✓ | ✓ | - | 52.7 | 36.7 | **2.5** | 335.7 | - | - | - |
| DnD (Li et al., 2022a) | ✓ | ✓ | - | 52.5 | 35.5 | - | 525.3 | - | - | - |
| NeurPhys (Shimada et al., 2021) | ✓ | ✓ | ✓ | 76.5 | 58.2 | - | - | - | - | - |
| SimPoE (Yuan et al., 2021) | ✓ | ✓ | ✓ | 56.7 | 41.6 | 6.7 | - | - | - | - |
| OSDCap (Le et al., 2024a) | ✓ | ✓ | ✓ | 54.8 | 39.8 | 8.4 | 132.8 | 119.1 | 16.0 | 15.2 |
| QuaMo_TRACE (Ours) | ✓ | ✓ | ✓ | 51.3±0.11 | 37.5±0.05 | 5.7±0.03 | 116.2±1.04 | 101.4±1.37 | 7.8±0.06 | 6.6±1.78 |
| QuaMo_HMR2.0 (Ours) | ✓ | ✓ | ✓ | **46.7**±0.04 | **30.6**±0.03 | 5.3±0.04 | **88.8**±0.21 | **78.5**±0.33 | **6.8**±0.07 | 4.3±0.04 |

Table 1: Quantitative results on the Human3.6M dataset (Ionescu et al., 2014). Tmpl.: Template-based approach (*i.e.* SMPL-based). Kin.: kinematics-based approach. Onl.: online approach. Online methods work with only one future target pose at each time step. **Bold** highlights the best results within the kinematics category. The proposed QuaMo reaches state-of-the-art performance on the MPJPE, P-MPJPE, G-MPJPE, and GRE with HMR2.0 as the meta-PD controller target. On the motion plausibly metrics Accel, G-Accel, FS, we consistently record better results compared to other online kinematics-based approaches.

The MPJPE calculates the average frame-wise L2 distance between estimated human joint 3D coordinates and the ground truth data. The second metric, P-MPJPE, is MPJPE with a rigid alignment between two poses. To evaluate the motion jitter, we consider the Accel metric (mm/frame$^2$), which measures the difference between the predicted joint acceleration and the ground truth. In addition, motion artifacts can only be observed in a world coordinate with global translation (Gärtner et al., 2022a). The G-MPJPE computes MPJPE in global coordinates without root alignment. We also compute the Global Root Error (GRE), similar to MPJPE, but only on root translation. The global jitter G-Accel is computed similarly to Accel without root alignment. Foot skating (FS) is measured as the percentage (%) of frames that contain foot movements more than 2cm during contact.

## 4.4 RESULTS

We report the quantitative evaluation results of QuaMo in Tab. 1, with comparison to five groups of related study: 1) keypoint-baed approaches that lifted from 2D keypoints (Peng et al., 2024; Sun et al., 2024; Lang & Chuah, 2025; Huang et al., 2025; Kim et al., 2025), 2) vision-based approaches that utilize temporal information (Kanazawa et al., 2019; Kocabas et al., 2020; Wan et al., 2021), 3) single-or-two-frame prediction only (Kanazawa et al., 2018; Tripathi et al., 2023; Li et al., 2021a; Sun et al., 2023; Goel et al., 2023), 4) kinematics-based methods that base their prediction on a large window of frames (Li et al., 2022a; Zhang et al., 2024) or trajectory-optimization methods (Shimada et al., 2020; Gärtner et al., 2022b;a; Xie et al., 2021), and most related to our work, 5) kinematics-based online methods that only consider two frames as input (Shimada et al., 2021; Yuan et al., 2021; Le et al., 2024a). The methods are ranked with respect to their performance on the MPJPE

| Data | Method | MPJPE | P-MPJPE | Accel | G-MPJPE | GRE | G-Accel | FS |
|---|---|---|---|---|---|---|---|---|
| Fit3D | TRACE (Sun et al., 2023) | 63.9 | 43.8 | 19.1 | 111.3 | 83.2 | 42.4 | 87.5 |
| | OSDCap (Le et al., 2024a) | 58.7 | 42.6 | 8.2 | 73.8 | 47.2 | 12.8 | 25.9 |
| | QuaMo_TRACE (Ours) | **50.3**±0.13 | **35.6**±0.04 | **3.8**±0.01 | **68.8**±0.21 | **45.2**±0.15 | **5.6**±0.03 | **16.3**±0.27 |
| Sports-Pose | TRACE (Sun et al., 2023) | 99.3 | 68.7 | 14.7 | 421.7 | 389.1 | 39.1 | 38.5 |
| | OSDCap (Le et al., 2024a) | 71.7 | 52.4 | 10.9 | 113.6 | 90.2 | 17.1 | 38.0 |
| | QuaMo_TRACE (Ours) | **71.4**±0.30 | **48.7**±0.21 | **5.3**±0.19 | **112.2**±0.75 | **82.3**±0.36 | **13.7**±0.13 | **24.1**±0.91 |
| AIST | TRACE (Sun et al., 2023) | 115.6 | 63.2 | 34.1 | 243.8 | 208.3 | 107.5 | 104.4 |
| | HUND (Zanfir et al., 2021) | 107.4 | 66.9 | - | 155.7 | - | - | 50.9 |
| | DiffPhy (Gärtner et al., 2022a) | 105.5 | 66.0 | - | 150.2 | - | - | 19.6 |
| | HMR2.0 (Goel et al., 2023) | 101.9 | 60.2 | 24.4 | 154.3 | 110.5 | 40.7 | 56.0 |
| | QuaMo_HMR2.0 (Ours) | **89.1**±0.14 | **60.0**±0.20 | **14.7**±0.02 | **144.1**±0.57 | **108.7**±1.07 | **14.9**±0.07 | **13.0**±0.46 |

Table 2: Quantitative results on the Fit3D (Fieraru et al., 2021) (top), SportsPose (Ingwersen et al., 2023) (middle) and the AIST (Li et al., 2021b) (bottom) dataset. Compared to OSDCap (Le et al., 2024a), QuaMo achieves a better performance on Fit3D and SportsPose, especially on the jittery metrics, using the same input TRACE. On AIST, with HMR2.0 as input, the proposed online QuaMo outperforms an offline method, DiffPhy, on both pose accuracy and motion jitter.

metric, within their respective category. Keypoint-based methods are only presented for referencing purposes and cannot be compared to template-based methods.

The proposed QuaMo is evaluated using two different online approaches: TRACE (Sun et al., 2023) and HMR2.0 (Goel et al., 2023) as the references for the PD controller. TRACE (Sun et al., 2023) directly regresses the 3D SMPL pose from input images, resulting in noisy 3D estimation with an Accel of 18.9 and FS of 80.3%. Using QuaMo, we manage to improve not only Accel and FS, but also MPJPE by 8.6%, P-MPJPE by 5.1%, and G-MPJPE by 18.7%. While TRACE is used for a fair comparison to competitors, we further improve the performance by using the newer HMR2.0 for the PD controller targets, which achieves state-of-the-art results on MPJPE, P-MPJPE, and G-MPJPE across all categories, while producing more plausible motions compared to other online methods. The improvement compared to HMR2.0 on Accel is 41.8%, 59.5% on G-Accel, and 62.6% in FS.

Our direct competitor is OSDCap (Le et al., 2024a), which is also an online dynamics-based approach. We outperform OSDCap on every metric, while using the same PD controller's target as TRACE (Sun et al., 2023): notably, 6.3% on MPJPE, 32.1% on Accel, and 12.5% on G-MPJPE. OSDCap, despite achieving a good state estimation through a Kalman-filter approach, re-introduces implausibility from the inputs obtained by TRACE back to the final output. We, however, achieve much smoother and plausible motions by fully respecting the temporal relationship between consecutive predictions from the integration scheme. DnD (Li et al., 2022a) and PhysPT (Zhang et al., 2024), despite having a competitive performance, take a window of 16 frames as input, while ours only takes one next frame as target. PhysPT (Zhang et al., 2024) achieves a smoother motion than our QuaMo; however, their motion is reconstructed from a seq2seq transformer model (Vaswani et al., 2017), without having an integration scheme to ensure temporal dependency between consecutive frame predictions. TrajOpt (Gärtner et al., 2022b) has a better FS measurement due to their offline trajectory optimization approach with a global refinement, whereas QuaMo is fully online and still maintains a competitive FS of 4.3% (compared to 4.0% of TrajOpt) with much lower MPJPE.

While the Human3.6M dataset serves as a baseline for benchmarking human pose estimation approaches, the variability of motions in the dataset is limited. Therefore, we also conduct an evaluation on the Fit3D (Fieraru et al., 2021) and SportsPose (Ingwersen et al., 2023) datasets with more complex and more challenging sports movements, presented in Tab. 2. Similar to the results in Tab. 1, we improve upon the input TRACE by a large margin and outperform OSDCap on all metrics. Additionally, we follow DiffPhy (Gärtner et al., 2022a) and evaluate QuaMo on the same subset of the AIST database (Li et al., 2021b), shown in Tab. 2. Because the implementation of HUND (Zanfir et al., 2021), the target input of DiffPhy, is not publicly available, we use HMR2.0 as our PD target instead. QuaMo achieves better results on all accuracy and plausibility metrics. Some qualitative results showing the advantage of QuaMo are presented in Fig. 3. Additional visualizations can be found in Supplementary B and videos on our supplemental webpage.

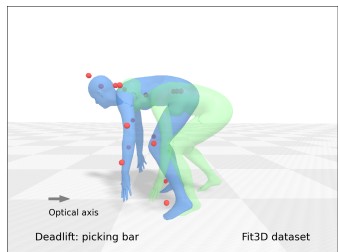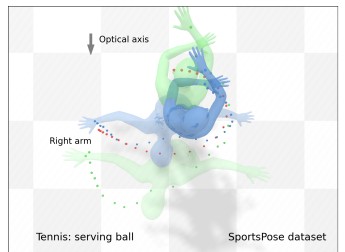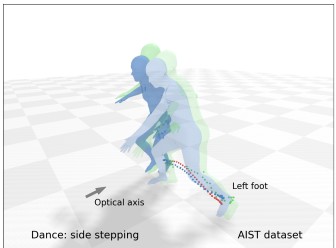

Figure 3: Qualitative results on three datasets: Fit3D (left), SportsPose (middle), AIST (right). QuaMo's predictions are shown in blue, the input (from TRACE or HMR2.0) in green, and ground truth keypoints in red for reference. The start frame has lower transparency. The reconstructed motions from QuaMo have significantly lower jitter and higher accuracy along the optical axis.

| Method | Rotation | $f_\omega$ | $\mathcal{S}^3$ | $\alpha$ | MPJPE | P-MPJPE | Accel | G-MPJPE | G-Accel | FS |
|---|---|---|---|---|---|---|---|---|---|---|
| TRACE | Axis-angle | | | | 56.3 | 39.3 | 19.3 | 146.4 | 44.5 | 80.3 |
| PD only | Euler XYZ | - | - | - | $74.4_{\pm 3.86}$ | $39.7_{\pm 0.84}$ | $13.7_{\pm 2.02}$ | $148.7_{\pm 3.71}$ | $15.4_{\pm 1.91}$ | $17.3_{\pm 2.38}$ |
| PD only | Euler ZXY | - | - | - | $71.2_{\pm 2.17}$ | $41.1_{\pm 0.71}$ | $7.6_{\pm 1.02}$ | $143.9_{\pm 2.76}$ | $9.3_{\pm 0.92}$ | $19.7_{\pm 2.18}$ |
| PD only | Axis-angle | - | - | - | $60.6_{\pm 1.40}$ | $39.0_{\pm 0.14}$ | $6.4_{\pm 0.99}$ | $137.8_{\pm 1.32}$ | $8.2_{\pm 0.93}$ | $17.2_{\pm 0.93}$ |
| PD only | Quaternion | - | - | - | $53.8_{\pm 0.18}$ | $38.8_{\pm 0.15}$ | $5.7_{\pm 0.07}$ | $132.6_{\pm 3.22}$ | $7.9_{\pm 0.94}$ | $19.5_{\pm 1.23}$ |
| QuaMo | Quaternion | ✓ | - | - | $53.1_{\pm 0.05}$ | $38.6_{\pm 0.04}$ | $5.3_{\pm 0.03}$ | $115.9_{\pm 0.08}$ | $8.6_{\pm 0.12}$ | $13.9_{\pm 0.57}$ |
| QuaMo | Quaternion | ✓ | ✓ | - | $52.0_{\pm 0.07}$ | $38.1_{\pm 0.05}$ | $\mathbf{5.2}_{\pm 0.02}$ | $114.8_{\pm 0.82}$ | $\mathbf{7.8}_{\pm 0.09}$ | $10.6_{\pm 0.30}$ |
| QuaMo | Quaternion | ✓ | ✓ | ✓ | $\mathbf{51.3}_{\pm 0.08}$ | $\mathbf{37.4}_{\pm 0.04}$ | $5.9_{\pm 0.02}$ | $\mathbf{114.7}_{\pm 1.01}$ | $8.4_{\pm 0.04}$ | $\mathbf{10.0}_{\pm 0.30}$ |

Table 3: Ablation studies. The baseline uses only a PD controller (PD only), taking TRACE as targets. The $f_\omega$: using the data-driven bias in Eq. 5; $\mathcal{S}^3$: using the integration method from Eq. 3 and Eq. 4; $\alpha$: using the acceleration enhancement term in Eq. 5. The model configurations are ranked based on their MPJPE score, and the lowest MPJPE with a reasonable Accel is most desired.

## 4.5 ABLATION STUDIES

We conduct two ablations in Tab. 3 to verify the usage of our proposals: using the quaternion as joint representation, and using the quaternion $\mathcal{S}^3$ constraint for integration and the acceleration enhancement. To reduce computational load, all ablations are conducted on a subset of Human3.6M taken from camera 60457274. Methods with lower MPJPE are more desirable. TRACE is chosen as the baseline and is presented in the first row of Tab. 3 for comparisons.

**Joint rotation.** We first compare the common joint representations: Axis-angle, Euler ZXY, Euler XYZ, and quaternion (ours), in row 2 to 5 in Tab. 3. As described in Sec. 1, axis-angle and Euler angles suffer from discontinuities over all three angles, causing instability during integration. An example is shown in Fig 4, when the temporally consistent human reconstruction has to make a full 180-degree rotation when the root joint encounters a discontinuity. This situation can be easily avoided by using quaternions. The experimental result of Euler angles in Tab. 3 shows that while having a reasonable MPJPE, the Accel is significantly larger, due to the constant compensation that the model has to produce to overcome the angle discontinuity. The axis-angle representation is more robust to the discontinuity; however, the error between two axis-angles for the PD controller cannot be defined in a physically meaningful way. We instead apply finite differences between each of the three components of the axis-angles separately as the error. As demonstrated with relevant metrics in Tab. 3, it is more difficult to correctly capture the motions with axis-angles than with quaternions.

**QuaMo's components.** We gradually add the proposed solution to the baseline with quaternion rotation. The data-driven $f_\omega$ helps address the database-specific offsets that ensure accurate estimation, especially the global translation with G-MPJPE decreases from 132.6 to 115.9 mm. The quaternion integration with the spherical constraint $\mathcal{S}^3$ reduces the error of the QDE solving process, leading to a decrement from 53.1 to 52.0 mm in MPJPE. The proposed acceleration enhancement term with its adaptive ability further improves the estimation accuracy, with an MPJPE decrease from 52.0 to 51.3 mm. A trade-off of the acceleration term is the motion jitter (Accel increases from 5.2 to 5.9) due to the noise amplification from the second-order differences of the input TRACE. Despite the jittery

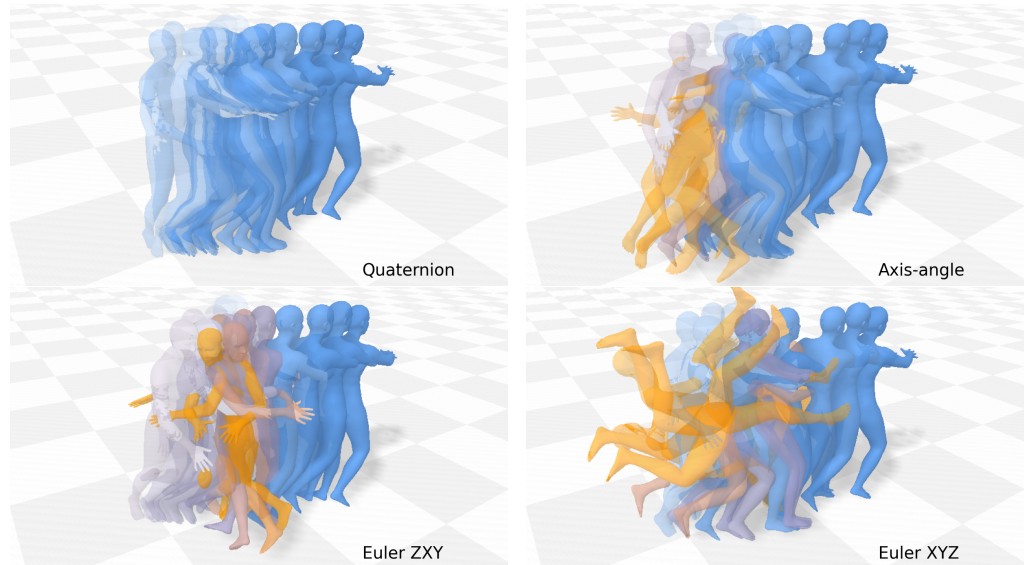

Figure 4: An example of motion reconstruction when a discontinuity occurs in the root joint rotation for different rotation representations. Blue means low and orange high MPJPE. The transparency corresponds to the time steps in the sequence. The model attempts to compensate for the discontinuity by rotating along the different rotation axes for all representations, except our quaternions.

trade-off, our proposed method enables accurate estimations for real-time downstream applications. The computational complexity of QuaMo is presented in Supplementary D.

Besides the training hyper-parameters taken from previous work OSDCap Le et al. (2024a) such as the same batch size $64$, learning rate $5e - 4$, hidden dimension $512$, and the usage of LayerNorm and LeakyReLU activations; we additionally provide an ablation study on the choice of shape loss scaling factor $\lambda$ in Tab 4. Because the Accel between different $\lambda$ are similar, we choose $\lambda = 0.01$ as our final selection due to the good trade-off between the local MPJPE and the global G-MPJPE.

| $\lambda$ | MPJPE | G-MPJPE | Accel |
|---|---|---|---|
| 1.0 | 52.8 | 120.2 | 5.8 |
| 0.1 | 51.8 | 118.6 | 5.8 |
| 0.01 | 51.2 | **116.2** | 5.7 |
| 0.001 | **51.0** | 116.9 | 5.7 |
| 0.0 | 51.1 | 117.1 | 5.7 |

Table 4: Ablation study on the shape loss scaling $\lambda$. We choose $\lambda = 0.01$ as our final selection due to the good performance trade-off between the local MPJPE and the global G-MPJPE.

## 5 CONCLUSION

In this paper, we introduce QuaMo, a novel online vision-based human kinematics capture method for recovering plausible human motions from cameras. Prior works often make use of Euler angles as the joint representation, which creates inaccurate solutions due to discontinuity during temporal integration. We propose the usage of the quaternion differential equation together with unit-sphere constrained solutions and acceleration enhancements for accurate and plausible 3D kinematics capture. We evaluate QuaMo in comparison with related work and achieve state-of-the-art results on four datasets: Human3.6M, Fit3D, SportsPose and AIST.

**Limitation and future work**. While human kinematics can be accurately estimated with QuaMo, the influence of the surrounding environment on the velocity predictions has the potential to further improve the plausibility of the reconstructed motions. As a natural extension, contacts and interactions of humans with external scenes will be investigated in future work.

ACKNOWLEDGMENTS AND DISCLOSURE OF FUNDING

This research is supported by the Wallenberg Artificial Intelligence, Autonomous Systems and Software Program (WASP), funded by Knut and Alice Wallenberg Foundation. The computational resources were provided by the National Academic Infrastructure for Supercomputing in Sweden (NAISS) at C3SE, and by the Berzelius resource, provided by the Knut and Alice Wallenberg Foundation at the National Supercomputer Centre. U.W. acknowledges funding from the Connected Minds Program, supported by Canada First Research Excellence Fund, Grant #CFREF-2022-00010.

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
