# OpenReview forum: "QuaMo: Quaternion Motions for Vision-based 3D Human Kinematics Capture"
_ICLR.cc/2026/Conference — ICLR 2026 Poster_

### Official Review · Reviewer_qvAF · 2025-10-25

**Soundness:** 3
**Presentation:** 3
**Contribution:** 3
**Rating:** 6
**Confidence:** 4

**Summary:**

This paper proposes QuaMo, an online kinematics module that replaces Euler-angle dynamics with a quaternion differential equation (QDE) solved exactly on the unit sphere $S^3$, coupled with a meta-PD controller augmented by a second-order acceleration enhancement term. Given per-frame vision priors (e.g., TRACE, HMR2.0), QuaMo predicts angular velocities and integrates quaternions via the Hamilton product to reduce the motion jitter and produce temporally consistent motion from the monocular videos.

**Strengths:**

- A key strength of QuaMo is its fully online state-space formulation, which does not rely on future observations, enabling real-time refinement of off-the-shelf 3D pose estimators through iterative updates of angular velocity and quaternion states.
- Unlike integration approximation methods, which risk moving the estimated quaternion outside the unit sphere $S^3$ and therefore require normalization to mitigate this issue, this work uses the Hamilton product between the quaternion representation of the rotation matrix and the current $q_t$. This leads to more accurate estimations, since normalization in previous approaches turns the quaternion into a direction that does not correspond to the true rotation and distorts the trajectory.
- The proposed acceleration enhancement $\alpha$ is computed from the second-order quaternion difference of the last three reference poses. It boosts control only when the reference is changing quickly (i.e., during true fast movement) and dampens as the target is reached. This allows QuaMo to treat fast, intended motion and jitter differently.
- For evaluation, the work reports both local and global metrics (MPJPE, P-MPJPE, Accel, global MPJPE/GRE, global jitter, and foot-skating) and provides error bars across random seeds. Evaluating the proposed QuaMo using TRACE and HMR2.0 as baselines, the results show that incorporating QuaMo significantly reduces acceleration errors. While the improvements in joint-based errors for HMR2.0 in camera coordinates are marginal, the method substantially decreases errors in the world coordinate system.

**Weaknesses:**

- As shown in table 3, although the proposed acceleration term $\alpha$ enhances the joint-based errors, it increases the acceleration error. It would be useful to clarify when to disable/attenuate the term to prevent the rise of acceleration error.
- Although the addition of Euler integration for root translation shows an improvement in GRE, it is a first-order numerical integration method, and I suspect that small errors could accumulate over time. A figure or analysis of translation error versus time would be useful.
-  There are minor ambiguities in the writing. First, $f_\omega$ is introduced as a function of $q_t$ and $\omega_t$. Then, at line~240, the term $b_t$ is introduced as an approximation of $f_\omega$, but $b_t$ is the output of the ControlNet, which receives additional inputs.
- Minor writing issues:
  - line 240 misses close parenthesis
  - line 236 has redundant comma after imaginary
  - line 290 mentions $\kappa_I$ of Eq. 5 that does not exist.
  - In table 1 and 2, Foot skating is mentioned as FS while in table 3 FK is used.

**Questions:**

- Given that the input $\hat{q}_t$ is noisy and the acceleration enhancement term uses a second-order difference, it can inherently amplify high-frequency noise. How is noise propagation controlled? Is it handled by ControlNet by assigning an appropriate $\kappa_A$ depending on the level of noise present in $\hat{q}_t$?
- Can the QuaMo method be applied to per-frame approaches that estimate MANO parameters, such as HaMeR [1]? If so, what modifications would be required to adapt the quaternion-based kinematic formulation to hand-specific articulations?

[1] Pavlakos, Georgios, Dandan Shan, Ilija Radosavovic, Angjoo Kanazawa, David Fouhey, and Jitendra Malik. "Reconstructing hands in 3d with transformers." In Proceedings of the IEEE/CVF Conference on Computer Vision and Pattern Recognition, pp. 9826-9836. 2024.

---

> ### Author Response · Authors · 2025-11-16
> **Response to Reviewer qvAF**
>
> Thank you for your recognitions and insightful feedback on our paper.
> We would like to further clarify the remaining concerns as in the following:
>
> - Acceleration enhancement term $\alpha$.
>
> The $\alpha$ term helps achieve a more accurate pose with a small jitter trade-off.
> The propagated noise is explicitly handled by two terms: the adaptive scaling of $\kappa_A$ and the derivative term scaled by $\kappa_D$.
> The ControlNet is trained to produce smooth predictions via the global consistency loss $\mathcal{L}_{global}$, making $\kappa_A$ adaptively change depending on the inputs.
> The derivative term additionally serves as a noise dampening for the PD controller, denoted by the negative sign in Eq.5, preventing the joint rotations to respond too quickly to noisy reference pose.
> In the future, a simple Kalman filter for adaptively weighting the contribution of the PD controller and the $\alpha$ term based on the noise level of the input signals could greatly benefit the method.
>
> - Euler integration of root trajectory.
>
> We added a root trajectory plot of QuaMo, similar to Fig.3 with a top-view with only the root joint, from a walking sequence taken from the Human3.6M test data, to the Supplementary materials.
> In QuaMo, we solve the ODE of the root translation from $t-1$ to $t$ via a single-step integration that suffers no numerical error.
> The equation in line 251 is solved with $\Delta t = 0.04$, which is the real-time transitioning of a motion capture sequence of 25Hz.
> Previous methods, such as NeurPhys, splits the integration into six iterations, causing the numerical errors from Euler's method to build up and accumulate over each iteration, while QuaMo does not have this problem.
>
> - Application to MANO.
>
> Extending QuaMo to work with MANO, or more generally the SMPL+H model, is very straightforward.
> The MANO hand model introduces an additional 30 parameters, consisting of 10 fingers with 3D axis-angle representation each.
> The process of transforming axis-angle to quaternion could be done similarly to the body poses in the original SMPL model.
> The ControlNet architecture could easily be extended in dimension, and all the computation of the PD controller remain the same.
> Extending QuaMo to also consider the dynamics influences from the fingers to the stability of captured motions is a very promising research, i.e. in climbinng sports, finger grips determine how effectively the whole body moves.
>
> - Manuscript typos.
>
> Thank you for your suggestions. We fixed and updated the manuscript accordingly.

---

> > ### Comment · Reviewer_qvAF · 2025-11-26
> >
> > Thank you for the response. My main concern was the issue regarding the root trajectory, and the newly added Figure 8 in the supplementary material clearly demonstrates that QuaMo does not suffer from drift over time. In fact, it consistently stabilizes and improves the baseline performance. In addition, QuaMo offers a notable contribution to online 3D human kinematics capture. I have raised my score.

---

> > > ### Author Response · Authors · 2025-11-26
> > >
> > > Thank you for considering our responses and for raising the score.

---

### Official Review · Reviewer_BjYh · 2025-10-29

**Soundness:** 3
**Presentation:** 3
**Contribution:** 2
**Rating:** 4
**Confidence:** 4

**Summary:**

The paper presents QuaMo, an online 3D human kinematics capture framework from monocular video. The method replaces Euler angles with a quaternion differential equation (QDE) under a unit-sphere constraint to avoid discontinuity and gimbal lock. It further combines a meta-PD controller with a second-order acceleration term to better handle fast motion changes. Experiments on four benchmarks (Human3.6M, Fit3D, SportsPose, AIST) indicate improved accuracy and smoother motion over several online kinematics baselines, with ablation results supporting each component.

**Strengths:**

Clear articulation of the discontinuity/gimbal lock problem and why Euler angles are problematic in online capture.

Correct quaternion-based formulation with integration respecting the unit-sphere constraint.

Adaptive acceleration term adds responsiveness to the PD controller, helping in fast motion regimes.

Solid empirical evaluation across multiple datasets, and thorough ablation on rotation representations and pipeline components.

**Weaknesses:**

Novelty: Quaternions for rotation representation are standard in robotics, graphics, and physics simulation. The specific QDE formulation and its integration here are sound, but not a fundamentally new concept.

No cost analysis: The paper doesn’t compare training/inference speed, memory usage, or computational overhead with Euler/axis-angle setups. The practical trade-off is unclear.

Input dependency: Performance varies greatly depending on upstream reference pose quality (TRACE vs HMR2.0). There is no robustness study under noisy or degraded inputs.

Evaluation scope: Benchmarks target relatively clean, single-person motions. Multi-person, occlusion-heavy, or contact-rich scenarios are not tested.

**Questions:**

1. What is the training/inference time cost relative to Euler or axis-angle versions? Does the quaternion formulation require higher compute/latency?

2. How does the method perform when reference poses are distorted or noisy? Any filtering or noise-robust adaptation tested?

3. Could the approach be extended to longer-horizon online settings (multiple future poses) without sacrificing latency?

4. Would the method benefit from incorporating environment contacts (as mentioned in future work) into the controller?

---

> ### Author Response · Authors · 2025-11-16
> **Response to Reviewer BjYh**
>
> Thank you very much for you thoughtful feedback on our paper.
> We would like to provide the answers to your questions in the following section:
>
> - On the novelty of QuaMo.
>
> QuaMo's innovation lies in the integration of a meta-PD controller to the exact estimation of quaternion derivative, which does not exist in the current literature in monocular online 3D human motion capture.
> Our QDE formulation does bring high performance gains in motion accuracy and results in significantly less jittery artifacts compared to related online methods.
> In our opinion, knowledge transfer in the form of successful implementation of a proven concept in one domain to another domain is indeed a contribution, in particular when it requires adaptation to a different pipeline.
>
> - Complexity compared to Euler and axis-angle versions.
>
> Using an NVIDIA A100, we additionally measure the average processing time and GPU memory allocation per pass of QuaMo for different rotation representations in the table below:
>
> | Criteria | Quaternion | Axis-angle | Euler XYZ | Euler ZXY |
> | -------- | ---------- | ---------- | --------- | --------- |
> | Processing time (ms) | 5.59 | 5.43 | 4.41 | 4.38 |
> | Memory allocated (MB) | 51.59 | 51.11 | 51.11 | 51.11 |
> | MPJPE (mm) | 51.3 | 60.6 | 71.2 | 74.4 |
>
> The quaternion rotation introduces a slight computational overhead due to one extra parameter in its 4D representation, compared to other 3D representations.
> However, this trade-off in complexity is very mininal compared to the significant gains in 3D pose accuracy that QuaMo provides, illustrated via the MPJPE metric (15.34\%).
> Compared to the axis-angle version with the closest performance, the overhead of QuaMo is only 0.16ms ($\approx$ 2.95\%) in processing time and 0.48MB ($\approx$0.94\%) of extra GPU memory allocation.
>
> - Dealing with noisy reference poses.
>
> In the paper, we intentionally choose a very noisy reference signal, TRACE, to demonstrate QuaMo's robustness to heavy noise, especially on the two datasets Fit3D (dark background) and SportsPose (captured from a phone).
> Unlike prior work, such as NeurPhys or OSDCap, QuaMo does not require a low-pass filter to preprocess the reference signals prior to the PD controller, yet still records significantly more plausible estimations with up to 32.1\% lower Accel and 51.3\% lower G-Accel than OSDCap.
> This is the direct result of integrating the exact quaternion calculation to the PD controller to obtain the correct joint angular velocity estimation, which was not possible in prior work.
> Please have a look at our supplementary videos in *index.html* for qualitative evaluations on the noise level of TRACE in Fit3D/SportsPose, and the significant improvements in motion quality that QuaMo provides, while still being an online single-frame method.
>
> - Extension to longer-horizon online settings.
>
> QuaMo is desiged to be a single-frame prediction method and extending it to work with a window of future frames is straightfoward.
> The idea of using a window of future frames for refining the reference signals has been addressed by a previous work DnD, by attentively regressing an optimal reference pose based on a window of 16 future frames.
> In this work, we keep the same single-frame setting as OSDCap for a fair comparison, and the longer-horizon settings will create no additional latency to QuaMo if applied.
>
> - Benefits of enviroment contacts.
>
> By integrating the influence of contacts, QuaMo would greatly increase its physical plausibility in 3D pose estimation, e.g. foot-skating artifacts would be reduced.
> The reason is that the environment exerts its influences to the human body via reaction forces at all contact points, and together with gravity,
> these reaction forces affect the joint rotational acceleration through Newtonian mechanics.
>
> - Scaling to scene-complex settings.
>
> In this study, we evaluate on the same experimental setups as prior work to verify the contributions of QuaMo.
> Extension to more complex scenarios, i.e. multi-person or scenes with objects, is a natural progression of the project and we are actively exploring in this direction as discussed in Sec.5.

---

> > ### Comment · Reviewer_BjYh · 2025-11-26
> >
> > Thank you for your response, which has addressed part of my concerns.
> > The task in this paper is "real-time", "SMPL-based", and "slightly physical" 3D human motion estimation.
> > In this context, combining a PD controller with quaternions could indeed be considered innovative.
> > Similar to Reviewer 6eiP, I also suggest that the authors incorporate relevant works from the past two years into the main text.

---

> > > ### Author Response · Authors · 2025-11-26
> > >
> > > Thank you for your additional suggestion. We have now included more recent literatures from 2024 to 2025 to Sec.2 related work, Sec.4 experiments, and Tab.1.
> > >
> > > Here is the list of papers that has been added to our manuscript:
> > >
> > > | Type | Method | Year |
> > > | ---- | ------ | ---- |
> > > | Keypoint | KTPFormer, CVPR | 2024 |
> > > | Keypoint | PoseAnchor, ICCV | 2025 |
> > > | Keypoint | PoseMamba, AAAI | 2025 |
> > > | Keypoint | Mambapose, AAAI | 2025 |
> > > | SMPL | MeshPose, CVPR | 2024 |
> > > | SMPL | TokenHMR, CVPR | 2024 |
> > > | SMPL | MultiHMR, ECCV | 2024 |
> > > | SMPL | CameraHMR, 3DV | 2025 |
> > >
> > > Keypoint-based methods are only for reference.
> > > The recent SMPL-based methods are similar to TRACE and HMR2.0, estimating 3D poses from input images and do not consider the temporal consistency between 3D estimations.
> > >
> > > We would highly apprecite more specific pointers to the parts of your concerns that have not yet been sufficiently addressed, so we could provide further clarifications.

---

> > > > ### Comment · Reviewer_BjYh · 2025-11-27
> > > >
> > > > Thank you for the author's further clarification. I will update my score after considering the responses from other reviewers.

---

### Official Review · Reviewer_6eiP · 2025-10-29

**Soundness:** 2
**Presentation:** 3
**Contribution:** 2
**Rating:** 4
**Confidence:** 3

**Summary:**

The paper targets the challenging problem of 3D human motion capture based on visual input. To address the problem of implausible and jittery motion due to the representation of Euler angles, the paper propose to use Quaternion motions instead and an effective system based on QDE  is utilized. The experiments on several benchmarks like Human3.6M, Fit3D, SportsPose and AIST validate the effectiveness of the proposed algorithm.

**Strengths:**

* The paper targets the challenging problem of 3D human motion capture based on visual input, which is of great importance to the industry applications.

* The idea of using quaternion differential equations for human kinematics capture is intersting.

* It reports reasonable results on several benchmarks like Human3.6M, Fit3D, SportsPose and AIST.

**Weaknesses:**

* It should include the recent references published in the recent two years. Currently, there is no reference published in 2025.

* For the experiments, there are several releated works which are not compared. For example, [R1] is referenced in the paper but not compared in Table 1. By checking the results report in [R1], it would have obviously better results compared with the prposed algorithm. Please involve more papers published in the recent two years (2024-2025) for comparisons.

[R1] Jihua Peng, Yanghong Zhou, and PY Mok. Ktpformer: Kinematics and trajectory prior knowledge-
enhanced transformer for 3d human pose estimation. In CVPR, pp. 1123–1132, 2024

* In Section 4.2, there are several implementation details reported in the paper. As there are many hyper-parameters defined in the paper, is there possible to provide more ablations on the setting of these hyper-parameters?

**Questions:**

Please address the questions raised in the weakness section. More specifically, please provide more comparisons in Table 1 to validate the effectiveness of the proposed algorithm against the state-of-the-art algorithms.

---

> ### Author Response · Authors · 2025-11-16
> **Response to Reviewer 6eiP**
>
> Thank you for your thoughtful feedback and recommendations.
> We would like to provide the answers to your questions in the following section:
>
> - More reference literature in 2025.
>
> Until the ICLR 2026 submission deadline, there was little published work on real-time 3D human motion capture from cameras.
> The method suggested by the reviewer, KTPFormer (published in 2024), belongs to a different research area of 2D-3D keypoint lifting.
> The methods in this area (e.g. VideoPose3D, PoseFormer or STCFormer) are not online and they require an input window of 2D keypoint frames of size 243 to predict one 3D keypoint pose, completely overlook the plausibility of the 3D estimations such as bone-length consistency between consecutive 3D poses.
> We found a recent method that also falls into this category: PoseAnchor, ICCV2025.
>
> In contrast, QuaMo belongs to the group of methods solving a different task: estimating the human 3D joint rotations directly from the input images, to control a template SMPL model that explicity ensures the human pose plausibility.
> The performance between keypoint-based and SMPL-based approaches is not directly comparable and this argument has been raised by many previous works: DiffPhy [A], DnD [B], PhysPT [C].
> Compared to all SMPL-based methods in Tab.1, including QuaMo, keypoint-based methods (even in earlier work) tend to produce lower MPJPE due to the freedom of the 3D keypoint-based predictions to move freely in space, heavily violating the human body constraints.
> Furthermore, due to the simplicity in the transfering of pose parameters between templates, SMPL-based approaches have much higher scalabilities to downstream tasks such as simulation or animation.
>
> We collect the results from the keypoint-based methods in the following table:
>
> | Method | Year |MPJPE | P-MPJPE |
> | ------ | ---- |----- | ------- |
> | VideoPose3D | 2019 | 46.8 | 36.5 |
> | PoseFormer | 2021 | 44.3 | 34.6 |
> | KTPFormer | 2024 | 40.1 | 31.9 |
> | PoseAnchor | 2025 | 40.3 | 32.1 |
>
> We hope this clarifies the reviewer's comment on references and are open to further suggestions on other recent relevant literature.
>
> - Ablations for training hyper-parameters.
>
> The majority of training hyper-parameters are selected following our direct competitor OSDCap for a fair comparison on real-time 3D human kinematics estimations, e.g. the same batch size 64, learning rate 5e-4, hidden dimension 512, and the usage of LayerNorm and LeakyReLU activations.
> The number of epochs and decay steps are determined based on the training learning curve to achieve full convergence.
> We provide additional ablation studies on the choice of shape loss scaling factor $\lambda$ as below:
>
> | Lambda | MPJPE | G-MPJPE | Accel |
> | ------ | ----- | ------- | ----- |
> | 1.0    | 52.8  | 120.2   | 5.8   |
> | 0.1    | 51.8  | 118.6   | 5.8   |
> | 0.01   | 51.2  | **116.2** | 5.7 |
> | 0.001  | **51.0** | 116.9 | 5.7  |
> | 0.0    | 51.1  | 117.1   | 5.7   |
>
> We choose $\lambda = 0.01$ as our final selection due to the good performance trade-off between the local MPJPE and the global G-MPJPE.
> We added this ablation study to the supplementary materials.
>
>
> We hope our rebuttal sufficiently addresses the reviewer's questions and clarifies the raised concerns.
>
>
>
>
> [A] Gartner et al., Differentiable Dynamics for Articulated 3d Human Motion Reconstruction, CVPR, 2022.
>
> [B] Li et al., Learning Human Dynamics from Dynamic Camera, ECCV, 2022.
>
> [C] Zhang et al., Physics-aware Pretrained Transformer for Estimating Human Dynamics from Monocular Videos, CVPR, 2024.

---

> > ### Comment · Reviewer_6eiP · 2025-11-26
> >
> > Thanks for the rebuttal. The rebuttal addressed part of my concerns in the previous round of review.

---

> > > ### Author Response · Authors · 2025-11-26
> > >
> > > Thank you for your reply. We have now included more recent literature (2024-2025) and corresponding discussions to Sec.2 related work, Tab.1, and in Sec.4 experiments.
> > >
> > > We present the list of collected paper in the table below:
> > > | Type | Method | Year |
> > > | ---- | ------ | ---- |
> > > | Keypoint | KTPFormer, CVPR | 2024 |
> > > | Keypoint | PoseAnchor, ICCV | 2025 |
> > > | Keypoint | PoseMamba, AAAI | 2025 |
> > > | Keypoint | Mambapose, AAAI | 2025 |
> > > | SMPL | MeshPose, CVPR | 2024 |
> > > | SMPL | TokenHMR, CVPR | 2024 |
> > > | SMPL | MultiHMR, ECCV | 2024 |
> > > | SMPL | CameraHMR, 3DV | 2025 |
> > >
> > > As discussed in the rebuttal, keypoint-based methods are not comparable to SMPL-based due to natural body constraints and only presented for reference purposes.
> > > Regarding the SMPL-based methods, we additionally found four more recent papers: MeshPose, TokenHMR, MultiHMR and CameraHMR.
> > > Similar to TRACE and HMR2.0, these methods estimate SMPL parameters from input images and do not consider the temporal consistency between 3D estimations.
> > >
> > > We would highly appreciate more specific pointers to the parts of your concerns that have not yet been addressed, so that we could further clarify them.

---

### Author Response · Authors · 2025-11-16
**General response**

First of all, we would like to express our gratitudes to all reviewers for their efforts on providing us insightful comments and feedback to our paper QuaMo.
We are glad that the contributions of QuaMo towards the task of online 3D human motion capture is recognized as stronng (qvAF), clear and correct (BjYh), and interesting (6eiP).
For further clarity about our paper, we provide answers and complimentary experiments as requested to each reviewer in their respective comment sections.

---

### Author Response · Authors · 2025-11-25
**Discussion period finishing soon**

Dear ACs and Reviewers,

As we enter the final phase of the discussion period, we notice that our responses have not yet been reviewed.
We would appreciate any updates from the reviewers and remain available for further discussions.

---

### Author Response · Authors · 2025-12-01
**Rebuttal phase summary**

We would like to thank the Area Chairs and Reviewers for their precious time and efforts in reviewing our paper, especially during the current challenging time in the field.
To aid the final assessment, we briefly summarize the author-reviewer discussion process with only the main points from the reviewers in the following:

- In online 3D human motion capture, the innovation of the proposed method (QuaMo) is recognized by the reviewers as: strong (qvAF), innovative (BjYh), and interesting (6eiP).

- Reviewer qvAF asked for further clarification about root translation, concerning the stability of the Euler integration. With the additional root-trajectory plots in Fig.8, we clarified that the root joint is updated with real-time transitioning in one step, preventing the numerical errors from accumulating over sub-steps as in previous approaches. Reviewer qvAF agreed and raised their score.

- Reviewer BjYh was concerned about the novelty, computational costs, and noisy input evaluations. 1) Regarding the novelty, we clarify that knowledge transfer in the form of successful implementation of a proven concept (quaternion) in one domain to another domain is indeed a contribution, and Reviewer BjYh has acknowledged this during the discussion. 2) On computation, we additionally provide a comparison between quaternion vs other 3D rotations. The trade-off is minimal with quaternions in QuaMo, providing a significant performance gain (15.34%) with only a slight overhead of 2.95% in processing time and 0.94% in GPU memory. 3) About noisy input, we clarified that QuaMo was intentionally verified with a very noisy input from TRACE, and we referred Reviewer BjYh to our supplementary videos to examine the noise level of TRACE and the great improvements that our method provides. Ultimately, Reviewer BjYh mentioned that they will update their score.

- Reviewer 6eiP is mainly concerned about the references in recent years (2024-2025) and more ablative studies on the training hyperparameters. 1) Regarding the references, we additionally collected 8 more papers and added them to the text, and the comparison Tab.1. Most of the recent work is not directly comparable due to the differences in human pose representations and model configuration (i.e. online vs offline), and only shown for reference purposes. 2) Regarding the request for more ablations, we clarified that we used the same training hyperparameters as previous work for a fair comparison, and we provide one additional ablation on the scaling of the shape loss function.

For further clarification, please visit the respective reviewer sections and our paper.
Once again, we are very grateful for the time and efforts from the Area Chairs and Reviewers during the reviewing process.

Best regards,

Authors

---

### Meta-Review · Area_Chair_S95C · 2026-01-06

**Summary:**

While the reviewers acknowledge the interest of the proposed method, they express concerns regarding the discussion of and comparison with recent work, the lack of some aspects related to evaluation (ablation study of hyper-parameters, computational/memory cost w.r.t. other representations, multi-person and occluded scenarios), the novelty of the method w.r.t. the robotics and graphics literature, and some specific technical aspects.

**Reviewer Concerns:**

The rebuttal thoroughly addressed the reviewers' comments, and all reviewers participated early in the discussion. One reviewer increased their score, and another mentioned that they would update it after considering the responses to the other reviewers. The AC believes that the main point that may not have been convincingly addressed is the comparison with key point-based methods requested by Reviewer 6eiP.

**Reviewer Scores:**

Reviewer qvAF increased their score to a 6, and it is entirely possible that Reviewer BjYh would have done the same. The AC is not convinced that Reviewer 6eiP would have also increased their score. This leaves the paper with an overall borderline score, without any major concerns but without any strong support.

---

### Decision · Program_Chairs · 2026-01-26

Accept (Poster)